# A Randomized Controlled Dietary Intervention Improved the Serum Lipid Signature towards a Less Atherogenic Profile in Patients with Rheumatoid Arthritis

**DOI:** 10.3390/metabo11090632

**Published:** 2021-09-17

**Authors:** Helen M. Lindqvist, Linnea Bärebring, Inger Gjertsson, Antti Jylhä, Reijo Laaksonen, Anna Winkvist, Mika Hilvo

**Affiliations:** 1Department of Internal Medicine and Clinical Nutrition, Institute of Medicine, Sahlgrenska Academy, University of Gothenburg, 40530 Gothenburg, Sweden; linnea.barebring@gu.se (L.B.); anna.winkvist@gu.se (A.W.); 2Department of Rheumatology and Inflammation Research, Institute of Medicine, Sahlgrenska Academy, University of Gothenburg, 40530 Gothenburg, Sweden; inger.gjertsson@gu.se; 3Zora Biosciences Oy, 02150 Espoo, Finland; antti.jylha@thl.fi (A.J.); reijo.laaksonen@zora.fi (R.L.); mika.hilvo@zora.fi (M.H.)

**Keywords:** lipidomics, rheumatoid arthritis, diet, CERT2, CVD-risk, ceramides, phospholipids, lipidome

## Abstract

Diet is a major modifiable risk factor for cardiovascular disease (CVD). One explanation for this is its effect on specific lipids. However, knowledge on how the lipidome is affected is limited. We aimed to investigate if diet can change the new ceramide- and phospholipid-based CVD risk score CERT2 and the serum lipidome towards a more favorable CVD signature. In a crossover trial (ADIRA), 50 patients with rheumatoid arthritis (RA) had 10 weeks of a Mediterranean-style diet intervention or a Western-style control diet and then switched diets after a 4-month wash-out-period. Five hundred and thirty-eight individual lipids were measured in serum by liquid chromatography-tandem mass spectrometry (LC-MS/MS). Lipid risk scores were analyzed by Wilcoxon signed-rank test or mixed model and lipidomic data with multivariate statistical methods. In the main analysis, including the 46 participants completing ≥1 diet period, there was no significant difference in CERT2 after the intervention compared with the control, although several CERT2 components were changed within periods. In addition, triacylglycerols, cholesteryl esters, phosphatidylcholines, alkylphosphatidylcholines and alkenylphosphatidylcholines had a healthier composition after the intervention compared to after the control diet. This trial indicates that certain dietary changes can improve the serum lipid signature towards a less atherogenic profile in patients with RA.

## 1. Introduction

Cardiovascular diseases (CVDs) are the leading cause of global mortality and a major contributor to disability. Rheumatoid arthritis (RA) is a chronic autoimmune disease. It is characterized by systemic inflammation and joint damage, leading to 30–50% increased risk of CVD events compared to the general population [1]. The recommendation for CVD risk reduction in RA is primarily anti-inflammatory treatment, but it is equally important to treat traditional CVD risk factors. The latest European League Against Rheumatism (EULAR) guidelines from 2017 included lifestyle recommendations including physical activity and Mediterranean diet, despite the lack of evidence from dietary intervention studies in patients with chronic inflammatory diseases [2].

Diet influences risk factors for CVD such as high body mass index (BMI), blood pressure and fasting plasma glucose as well as dyslipidemia. Recently, bioactive lipids have also been discovered as independent risk factors for CVD. Concentrations of specific ceramides in serum have been shown to predict major cardiovascular events [3,4,5], and ratios of ceramides to associate with CVD death, independent of other risk factors [6,7,8]. For example, CERT2, a new risk score based on specific phosphatidylcholines and ceramides in serum, is a significant predictor of CVD mortality, as shown in several independent cohorts [9,10,11]. In addition, Ottosson et al. reported that a lipid signature (i.e., a pattern of specific bioactive lipids) can predict CVD up to 20 years prior to disease onset, independent of traditional cardiovascular risk factors [8].

Since dietary intake is complex, the associations between diet and serum lipid signature are still largely unexplored [3]. A recent study found that a vegetarian diet compared to a diet including meat particularly influenced phosphatidylcholines, lysophosphatidylcholines, and sphingomyelins (SM) in plasma [3]. Though in vitro and animal studies show that n-3 long chain polyunsaturated fatty acids (LC-PUFA) eicosapentaenoic acid (EPA) and docosahexaenoic acid (DHA) positively influence bioactive lipid, such as ceramide concentrations, findings from three human interventions were inconclusive [11,12,13]. In addition, previous studies have mainly studied the impact of diet on classes of lipids, such as triacylglycerols (TAGs), treated as one component. Recent lipidomic studies highlight that not all lipid species within one lipid class have the same effects [14]. This emphasizes the importance of investigating the effects of individual lipids instead of lipid classes to scrutinize their potential role in CVD development.

In summary, recent findings have shown that bioactive lipids are associated with CVD risk. Diet has the potential to influence these lipid species, but the evidence is scarce and contradictory. The main aim of the present work was to investigate if a proposed anti-inflammatory diet could decrease the CVD risk score CERT2 or other scores suggested to be associated with CVD risk in patients with RA. The second aim was to explore effects of the dietary intervention on lipid signature and specific lipid species.

## 2. Results

### 2.1. Subjects and Compliance

A detailed description of recruitment, subjects and compliance have been reported previously [15]. In this study, 36 women and 14 men were included and 47 completed at least one diet period and 44 completed both diet periods. Due to one missing sample only 46 are included in the main analysis of this paper (Figure 1). Most, about 60%, of the participants were overweight and among these, half were obese (BMI ≥ 30). Central obesity (waist >80 cm for women and >94 cm for men) was also prevalent; 70% in men and 75% in women. About one fifth of the participants had elevated fasting TAG concentrations (>1.7 mmol/L, according to European guidelines [16]) at baseline (Table 1), and half of the participants were taking at least one cardiovascular agent [17] (Table 1).

The compliance to the diets was regarded as good in 91% of the diet periods and there were no significant changes in weight. The intervention diet compared to the control diet led to a significantly lower intake of protein, zinc and saturated fat and a significantly higher intake of fiber, PUFA, vitamin D and selenium. In addition, intake of EPA and DHA were considerably higher during the intervention diet [15]. Compliance was also confirmed by changes in plasma phospholipid fatty acids [17], where PUFA and especially EPA and DHA were higher after the intervention diet compared to control diet and shorter saturated fatty acids, such as 14:0 and 16:0, were lower. A detailed description of the effects on blood lipids have been reported previously [17]. In short, the intervention diet in comparison to the control diet led to lower TAG, lower apolipoprotein B100, and higher HDL. The higher HDL was mainly causes by an increase of HDL in the lower density range.

### 2.2. Effect of Intervention on Lipid Composite Scores Associated with CVD Risk

The CERT2 risk score did not change significantly when comparing the two diets. Still, it increased during the control diet indicating that a diet high in saturated fat and low in PUFA has a negative influence on CERT2 (Table 2). Among the components of the score, only Cer(d18:1/18:0)/PC 14:0/22:6 was significantly lower after the intervention compared to after the control diet. This was explained by both lower Cer(d18:1/18:0) and higher PC 14:0/22:6 after the intervention diet compared to after the control diet. Almost all included lipids were improved by the intervention diet compared to the control diet; exceptions were Cer(d18:1/24:1) that did not change and PC 16:0/22:5 that increased during the control diet and decreased during the intervention diet. The sensitivity analysis resulted in the same statistical significances except for that PC 14:0/22:6 was not significant different between the periods, although the increase in the intervention diet was still significant (*p* = 0.011) (Table 2).

In addition, four component scores, long TAG score, CE score, PC score and short TAG score were evaluated. The first two scores were higher after the intervention diet compared to after the control diet whereas the last two scores were lower, indicating that the lipid composition improved (Figure 2). Changes in concentrations of lipid classes and scores (median and interquartile range) are presented in Appendix A Appendix A. CE score was no longer statistically significant in the sensitivity analysis.

### 2.3. Influence of the Dietary Intervention on Lipid Classes

In univariate analysis of lipid classes, TAG, phosphatidylglycerol (PG), alkenylphosphatidylcholine (PC(P)), alkylphosphatidylcholine (PC(O)), PC, lysoalkenylphosphatidylcholine (LPC(P)) and ceramides were significantly lower after the intervention compared to the control diet. For TAG, PCO and LPC, this was caused by a decrease during the intervention diet and for PC(P), LPC(P) and ceramides this was caused by an increase during the control diet. The sensitivity analysis yielded a similar result with the only difference that total TAG and LPC(P) were no longer significantly different when comparing the two diet periods (Appendix A Appendix A).

### 2.4. Influence of Dietary Intervention on Lipid Signature

OPLS-EP and OPLS-DA models were used to investigate discrimination between diet regimens and to identify variables responsible for class separation (Table 3). Permutation test confirmed that the OPLS-DA models were of high quality (Appendix A Appendix A).

It was possible to classify 100% of the control samples (post–pre) and 98% of the intervention samples (post–pre) correctly when comparing the lipid changes during the two periods (Table 3, Figure 3). In the sensitivity analysis, 86 and 91% were correctly classified, respectively, probably because of fewer samples to build the model on. The results indicate that the diets had a substantial influence on the lipid signature. 

### 2.5. Associations between Changes in Known Clinical Markers and Plasma Phospholipid Fatty Acids and Changes in Lipid Signature

The OPLS-model based on delta-values including clinical markers resulted in a model including TAG, total cholesterol, LDL and HDL and the model including plasma concentrations of fatty acids included 11 fatty acids (14:0, 16:0, 16:1 n-7, 18:1 n-7, 18:1 n-9, 18:2 n-6, 18:3 n-6, 20:3 n-6, 20:4 n-6, 20:5 n-3 and 22:6 n-6) that were significantly associated to the changes in lipid signature (Appendix A Appendix A).

### 2.6. Discriminating Lipid Species

The models revealed that most lipids contributed to the models. Nevertheless, the lipids that contributed the most to the separation in the models are presented in Table 4 and Figure 4 and were either plasmalogens or sphingolipids. All selected lipids were significantly different between the two diet regimens and decreased/increased within the diet periods (Table 4). The two sphingolipids glucosyl/galactosyl ceramide (Glc/GalCer) (d18:1/26:1= 44:2) and SM 44:3 increased during the intervention diet, while the phosphatidylcholines decreased. A common feature for the sphingolipids was that they contained very long chain lipids because the sum of the length of their two chains were 44 carbons compared to the PC where the sum of chain lengths were 38-40 carbons. Lipids that contributed to the separation between the diets but did not cluster with their class are pointed out in Figure 4. This was the case for TAG 18:1/18:1/22:6, cer 18:1/26:1 and 18:2/26:1, SM 44:2 and 44:3, and PC 40:2, 40:3 and 40:8.

### 2.7. General Factors Influencing the Lipid Signature

In the PCA model on baseline values, describing the relations among lipid species, the first component (17.3% of the explained variation (R^2^X)) was mainly driven by TAGs, diacylglycerols (DAG) and phosphatidylethanolamines (PE) versus cholesteryl esters (CEs), lysophosphatidylcholines (LPCs), LPC(P)s, lysoalkylphosphatidylcholine (LPC(O)) and PC(P)s. The second component (16.2% of the explained variation (R^2^X)) was mainly driven by Glc/Gal/Cer and acylcarnitine (AC) versus phosphatidylinositol (PI) and phosphatidylcholine (PC). On the other hand, not all lipids of a specific lipid group clustered and this depended on the fatty acid chain lengths. The TAGs driving the first component all included the shorter chain lengths and more saturated species. The patterns of the PCA therefore show that not all lipids form one class cluster. Most obviously, TAG containing more saturated and shorter fatty acids cluster and associate with a higher total triglyceride content. The OPLS-model on baseline values only included age. Sex, BMI, CRP and DAS28-ESR did not influence the model. An OPLS-model including reported dietary intake resulted in a model that included only fiber and PUFA. High fiber and PUFA were inversely related to shorter and more saturated TAG.

## 3. Discussion

This study shows that among patients with RA a healthy Mediterranean style diet led to a beneficial serum lipid signature associated with reduced risk for CVD, compared to a Western control diet. However, only one component in the CERT2 score was significantly improved, i.e., Cer(d18:1/18:0)/PC(14:0/22:6), and consequently the composite score CERT2 was unaltered.

However, all the ceramides included in the CERT2-score were lower after the intervention diet compared to after the control diet. Similar effects of n-3 LC PUFA on ceramides have been reported in in vitro studies, but the findings from human interventions are less conclusive [18]. In a randomized controlled parallel study results suggested that consuming 500 g/week of fatty fish increased plasma EPA and DHA concentrations and lowered total plasma Cer concentrations after eight weeks, compared to consuming lean fish or no fish [12]. Likewise, a vegetarian diet compared to a diet including meat led to a decrease in Cer(18:1/16:0). On the other hand, there were no changes in plasma Cer concentrations from an intervention with high dose of fish oil for 3 weeks [11], or a Mediterranean diet supplemented with extra-virgin olive oil or nuts for one year (PREDIMED) [13]. In the fish oil study, the short time of 3 weeks could explain the lack of effect; otherwise, this was a well-controlled study with a specified background diet. The compliance or plasma fatty acids after the Mediterranean diet in PREDIMED were not reported and consequently there might have been limited changes in the fatty acid intake explaining the lack of effect.

Many of the changes in lipid species concentrations in the presented study can be explained by an overall change in the concentration of the included fatty acids in those specific species. For example, EPA (20:5) and DHA (22:6) in plasma phospholipids were higher and stearic acid (16:0) was lower after the intervention diet compared to the control diet [17]. The changes in fatty acids can thus explain the increase in PC 16:0/22:5 during the control diet and decrease during the intervention diet as well as the changes in concentrations of PC 14:0/22:6. However, total ceramide concentrations have also been reported to increase in plasma after intake of saturated fat because of up-regulated genes involved in ceramide synthesis [18]. We can therefore not exclude that our findings are a result of the lower content of SFA in the presented intervention diet compared to control diet. However, our result suggests that the overall dietary fat composition influence the ceramide content in serum strongly.

The OPLS-DA and EP models indicate that most lipids were influenced by dietary changes and the selection of the most discriminating lipids must therefore be interpreted with caution since many more lipids had a substantial influence on the model. However, the lipid classes and species most affected by diet will be discussed in the following section.

Total TAG decreased during the intervention diet and increased during the control diet, but besides that the composition of TAGs fatty acids changed towards a content of longer fatty acyl side chains instead of shorter. Similar result has been reported earlier, since Lankinen et al. found that contents of EPA and DHA strongly correlated with the plasma long chain TAG [12] and Djecek et al. found that a vegetarian diet compared to a meat diet changed the TAG composition towards TAG with longer fatty acids [3]. Unfortunately, many lipidomics platforms include only half the number of lipids compared to our, which limits the possibility to compare the findings for most of the lipid species.

The class of PC(O):s decreased during the intervention diet and this was in agreement of the findings from the multivariate models where PC(O) 34:0, 38:3, 38:4b, 38:4c and 40:4 were among the most discriminant lipids. In addition, the increase of PC(P):s during the control diet was supported by the PC(P) 38:3, 40:3 and 40:6 from the discriminating lipids in the multivariate models comparing the diets. PC(O):s were also among the most discriminating lipids when comparing a vegetarian and a diet including meat [3]. It has been reported that PC(O) and PC(P) species display negative associations with area under the curve of insulin, especially in men, after an oral glucose tolerance test [19] and that PC(O) is associated to prediabetes and diabetes when compared to individuals with normal glucose test [20]. In addition, serum concentrations of PC(O) and PC(P) lipids were reported to have a significant inverse association with incident myocardial infarction (MI) in patients with mixed peripheral artery disease presentations [21] and specifically PC(P-40:6) had a strong inverse association with MI. However, we found an opposite effect on the lipid species PC(P-40:6), i.e., it was higher after the intervention diet compared to after control diet. Interestingly, Mundra et al. also investigated the associations between lipid species and MI in data from two large cohorts; the LIPID subcohort and the ADVANCE case cohort [5]. In their predictive model the selected lipid species followed the same pattern except for PC(P-40:6) that had opposing associations to MI incidence and death in the two cohorts. It had a stronger positive association in the non-diabetic participants. PC(P-40:6) was the only lipid species that had a significant interaction with diabetes of the selected lipids in their model. These contradictive findings remain to be explored, but an explanation could be that the fatty acids included in PC(P-40:6) could differ between the studies. For example, the increase of DHA (22:6) during the intervention diet in the presented trial suggests that an increase of PC(P-18:0/22:6) could be the cause for an increase in total PC(P-40:6) and that an increase in this lipids does not necessary have to be associated with negative health effects.

The two sphingolipids Glc/GalCer (d18:1/26:1) and SM 44:3 increased during the intervention diet, while phosphatidylcholines decreased. Similar results have been found previously after 3 weeks of fish oil compared to sunflower oil [11]. It is likely that SM (d18:1/26:2) composes at least a part of SM(44:3), indicating again that the intake of EPA and DHA is reflected in a wide range of lipid classes.

In the PREDIMED trial, short TAG-score, i.e., TAG:s with shorter fatty acids, at baseline was associated with increased risk for CVD, after adjustment for age, sex, BMI, family history of CVD, leisure-time physical activity and intervention group [22]. This indicates that our findings could have clinical relevance and a CVD-risk reducing effect. This is also reinforced by the increase of PC and CE-score during the intervention diet compared to control diet. These scores were also significantly associated with lower risk for CVD in the PREDIMED trial [22].

The OPLS models showed associations between lipid signature and age, which was expected. The OPLS including data from all samples (pre- and post- both diets) as well as dietary components only found associations to fiber and PUFA. This could be a consequence of that fiber and PUFA had the most pronounced changes in the intervention. Similarly, the OPLS model including clinical markers included TAG, total cholesterol, LDL, and HDL that all were influenced by the diet intervention although only changes in TAG and HDL were significantly changed.

The current trial has several strengths. First, the lipidomics analysis include a large number of lipid species, i.e., not only classes and in many cases even specified fatty acids which seems to be important to understand the effects of diet on the lipidome. Secondly, participants were instructed to keep a stable weight and did so, which is important because weight has a substantial effect on blood lipids. Third, the cross-over design control minimizes the effects of inter-individual variation, which is important for a group such as patients with RA where disease activity and pharmacological treatment exert effects on metabolism. As a consequence of the increased risk for CVD in patients with RA, many patients were on pharmacological treatment that influence the lipids directly, such as statins, or indirectly such as glucocorticoids and DMARDs. Despite this, the diet had a positive effect on the lipid signature. The sensitivity analysis excluding data from patients with changes in pharmacological treatment also confirmed our results. Lastly, the dietary data collected during both periods to control for compliance and to characterize the dietary intake as a whole, had a high quality. A limitation of the study is the generalizability because of the selected patient group; hence similar studies on healthy normal weight participants without any pharmacological treatment that might influence the lipids are required.

## 4. Materials and Methods

### 4.1. Study Participants and Study Design

Recruitment and dietary intervention took place during 2017–2018. Participating patients and study design were described in detail previously [15]. Briefly, an invitation was sent to all patients in the Swedish Rheumatology Quality Register (SRQ) with established RA (disease duration ≥2 years), 18–75 years old, and living in the Gothenburg region in Sweden. We included 50 women and men, with an active disease i.e., Disease Activity Score 28 (DAS28 ≥ 2.6) and they were allocated by computer-generated randomization to start with either an intervention or a control diet in a single-blinded cross-over intervention. After 10 weeks of either intervention diet or control diet and a wash out period of 2–5 months, participants switched diet regimen. The food was delivered weekly to the participants homes by a home delivery food chain. They received food to prepare breakfast, one snack, and one main dish per day for 5 day/week. In an attempt to blind participants, study staff referred to the intervention diet as “fiber diet” and the control diet as “protein diet” in all communication with participants, without mentioning which was hypothesized to be favorable. At enrolment, the participants were instructed to abstain from nutritional supplements other than those prescribed by a physician and to keep weight stable throughout the study.

All procedures were conducted according to the Declaration of Helsinki and approved by Gothenburg Regional Ethical Review Board (registration number 976-16, November 2016, and supplement T519-17, June 2017). Written and informed consent was provided by all participants.

### 4.2. Diets

The intervention diet was a diet combining foods with suggested anti-inflammatory effects, e.g., a Mediterranean style diet and the control diet intended to correspond nutritionally to the average dietary intake in men and women in Sweden (i.e., a Western diet). The diets have been described in detail elsewhere [15]. In brief, the intervention breakfasts contained wholegrain cereals, low-fat dairy, nuts, blueberries and pomegranate, and juice shots with probiotics. The main meals contained fish (mainly salmon) 3–4 times/week and vegetarian dishes 1–2 times/week. Two fruits per day were provided to eat between meals. The participants were also instructed to limit their intake of red meat to ≤3 times/week, to choose low-fat dairy and wholegrain cereals, to eat ≥5 portions/weekof fruit, berries, and vegetables (including those provided) and to use oil or margarine for cooking.

The control diet provided breakfasts based on either a mix of quark and yoghurt with corn flakes or white bread with a butter-based spread and cheese, and orange juice. The main meals contained red meat or chicken and mainly refined grain products, a low content of vegetables. For the meals not provided, the participants were instructed to consume meat ≥ 5 times/week, use butter for cooking and choose high-fat dairy, consume ≤ 5 portions/weekof fruit, berries, and vegetables and ≤1 serving of seafooday/week and to avoid products with probiotics. Compliance was assessed mid-diet during both periods, by a telephone interview [15]. Participants were asked to rate their consumption of provided foods from none (0 point), part of (1 point) or all (2 points) for each meal during the preceding week. Hence, for 5 days the maximum compliance score was 30. Participants scoring at least 80% (24 points) were regarded as compliant.

### 4.3. Dietary Assessment

The participants completed a 3-d weighed food record during three consecutive days before and after each period. A dietitian instructed participants to weigh all items consumed or if unable to do so, estimate the amounts with help from pictures. The participants were also to note details such as type of fish, vegetables or fat content. The food records were all analyzed in Dietist Net Pro version 18.12.16 (Kost och Näringsdata AB, Bromma, Sweden) by the same dietitian.

### 4.4. Data Collection and Laboratory Analysis of Descriptives

At screening, height was measured to the closest 0.5 cm with a wall-mounted stadiometer, without shoes. Before and after each dietary period, weight and blood pressure were measured and fasting blood samples were collected by venipuncture.

A specially trained research nurse at the Clinical Rheumatology Research Centre, Clinic of Rheumatology, blinded to the treatment, performed the joint examinations. The composite scores DAS28 with erythrocyte sediment rate (ESR) and DAS28 with C-reactive protein (CRP) include the numbers of tender and swollen joints out of 28 joints, the patient’s estimation of his/her general health on a visual analog scale (VAS-GH), and either ESR or CRP [23]. Body composition was measured by bioimpedance spectroscopy by ImpediMed SFB7 after 5 min in supine position. Adhesive electrodes were placed on the patient’s right foot and hand according to manufacturer’s instructions. FM and FFM were calculated with the software BioImp version 5.5.0.1 by the manufacturer’s equations. Participants reported any changes in medication or health care visits during the dietary periods.

Blood samples were immediately transported to Clinical Chemistry at Sahlgrenska University Hospital for routine analysis of ESR, CRP, serum total cholesterol, high density lipoprotein (HDL), LDL and TAG (Cobas 8000, Roche Diagnostica, Scandinavia AB, Solna, Sweden). Fatty acid analysis in plasma phospholipids was performed by gas chromatography and 21 fatty acids were quantified as previously described elsewhere [24].

### 4.5. Lipidomic Profiling

Fasting blood samples were collected in 5 mL vacutainer tubes (VACUETTE^®^, Kremsmünster, Austria, serum gel, 5 mL), turned approximately 5 times, and allowed to clot for 30 min at room temperature. Tubes were centrifuged at 2200× *g* for 10 min. The serum aliquots were kept at 8 °C for a maximum of 3 h before storage at −80 °C until analysis.

Lipidomic profiling was performed at Zora Biosciences Oy (Espoo, Finland). Lipid extraction was based on a method that has been described before [25]. In brief, 10 µL of 10 mM 2,6-di-tert-butyl-4-methylphenol (BHT) in methanol was added to 10 μL of sample. After that 20 µL of internal standards (Avanti Polar Lipids Inc., Alabaster, AL, USA) and 300 µL of chloroform:methanol (2:1, *v*/*v*) (Sigma-Aldrich GmbH, Steinheim, Germany) was added and the samples mixed and sonicated in a water bath for 10 min, followed by a 40 min incubation. The samples were centrifuged at 5700× *g* for 15 min and the upper phase was transferred and evaporated under nitrogen. Extracted lipids were resuspended in 100 µL of water saturated butanol and sonicated in a water bath for 5 min and 100 µL of methanol was added. The extracts were then centrifuged at 3500× *g* for 5 min, and finally the supernatants were transferred to the analysis plate for mass spectrometric (MS) analysis.

MS analyses have also been described in detail before [26]. In short, the analyses were executed on a hybrid triple quadrupole/linear ion trap mass spectrometer (QTRAP 5500, AB Sciex, Concord, ON, Canada) equipped with ultra-high-performance liquid chromatography (UHPLC) (Nexera-X2, Shimadzu, Kyoto, Japan). Chromatographic separation of the lipidomic screening platform was executed on an Acquity BEH C18, 2.1 × 50 mm id. 1.7 µm column (Waters Corporation, Milford, MA, USA). A scheduled multiple reaction monitoring algorithm were used to collect data and it was processed by Analyst and MultiQuant 3.0 software (AB Sciex). Internal standards of the lipid classes were used to normalize the heights of the peaks obtained from MS analysis. 538 individual lipid species were identified.

### 4.6. Lipid Score Calculation

The main aim of this study was to analyze the composite risk score CERT2. CERT2 consists of three lipid ratios (Cer(d18:1/24:1)/Cer(d18:1/24:0), Cer(d18:1/16:0)/PC 16:0/22:5, Cer(d18:1/18:0)/PC 14:0/22:6) and a single lipid concentration (PC 16:0/16:0), and its calculation has been described in detail previously [6]. The cut-off values for score calculation were based on the Finnish FINRISK 2002 primary prevention study [27]. For comparison, the following scores were also calculated: Phosphatidylcholine (PC) score (the sum of PC, lysophosphatidylcholine (LPC) and alkylphosphatidylcholine (PC(P)) with ≥5 of double bonds), Cholesteryl ester (CE) score (sum of CE >4 double bonds), Long TAG score (sum of TAG > 54 carbons and ≥5 double bonds) and Short TAG score (sum of TAG <50 carbons and ≤4 double bonds) that all have been associated with CVD risk [22].

### 4.7. Multivariate Statistical Methods

All multivariate analyses were executed with SIMCA software v.15.0 (Umetrics AB, Umeå, Sweden) and the main analysis included 180 samples from 46 individuals who completed at least one diet period. Separation of classes and variables related to separation in the data according to classification of diet (*intervention* vs. *control*) were evaluated by an Orthogonal Projections to Latent Structures with effect projections (OPLS-EP), where delta values between periods (post intervention-post control) were used because the samples were paired (*n* = 43 subjects). In addition, OPLS with Discriminant Analysis (OPLS-DA), where delta values for each period (post–pre) were used (*n* = 90 periods), was performed. Cross-validation groups were set to the number of study participants and were based on individual identity, so that all samples from one individual were left out in one cross-validation round. The validity of OPLS-DA models was assessed by permutation tests (*n* = 999). If lipids were among thirty highest VIP-scores or w > 0.09, class discriminating variables of interest from the OPLS-EP and OPLS-DA models were selected.

Principal component analysis (PCA) models and Orthogonal Projections to Latent Structures (OPLS) were used to explore clustering patterns of observations and trends in the data in relation to known factors and outliers. OPLS models include not only x-variables, such as in this study lipid data, but also y-variables, i.e., additional known factors that could influence the data. Sex, age, disease severity, body mass index (BMI) and CRP were tested as y-values in a baseline OPLS-model. Dietary intake i.e., energy, carbohydrates, proteins, fat, SFA, MUFA, PUFA and fiber were tested as y-values in an OPLS-model including all samples. To further explore the changes in the lipid signature caused by diet, delta-values for both dietary periods were tested in one OPLS-models with the clinical markers; ESR, CRP, TAG, total cholesterol, LDL and HDL and in another OPLS-model with the fatty acids concentration in plasma phospholipids as y-variables. Included fatty acids have been reported previously [17]. For all presented OPLS-models only y-variables that have a CV-ANOVA *p* < 0.05 for that specific model are included.

### 4.8. Univariate Statistical Methods

Statistical analyses were performed with SPSS version 25 (SPSS Inc., Chicago, IL, USA). The main analysis for CERT2 was performed by Wilcoxon signed rank test. Treatment effect of included quotes and lipids in CERT2, lipid groups and selected discriminating lipids were evaluated by a mixed model with treatment (intervention or control), period and sequence as fixed effects and subject nested in sequence as a random effect. Baseline values for the outcome measure were included as a covariate. Because several lipids were non-normally distributed, all data was natural logarithm transformed for the mixed model analysis. Wilcoxon signed rank test was used to compare pre and post values within each diet.

To compare effects on lipid classes and scores, delta values for each dietary period (post–pre) were Z-score standardized and presented as mean and 95% CI. In this explorative study significance was set at α = 0.05, i.e., not corrected for multiple hypothesis testing.

The main model included all data from individuals who completed at least one diet period. Sensitivity analysis included participants who fulfilled an a priori set of requirements. In these analyses, only those participants who completed the whole trial, had good compliance in both diet periods (≥80%), and without changes in use of statins or glucocorticoids during the whole study period were included (*n* = 28).

### 4.9. Power Calculation

The power calculation was based on expected changes in the primary outcome of the trial, DAS28-ESR. To detect a change of 0.6 units in DAS28-ESR with 90% power and α = 0.05, a sample size of 38 patients was needed.

## 5. Conclusions

This trial indicates beneficial effects on the serum lipid signature in patients with RA after a Mediterranean like intervention diet low in meat and rich in fatty fish, whole grain, fruit, vegetables, nuts, and including low fat dairy products, compared to a Western diet. Beneficial effects were seen on concentrations of ceramides, alkylphosphatidylcholines, alkenylphosphatidylcholines and a healthier composition of triacylglycerides, cholesteryl esters, and phosphatidylcholines after the intervention diet compared to after control diet. However, there was no improvement in the CERT2 risk score.

## Figures and Tables

**Figure 1 metabolites-11-00632-f001:**
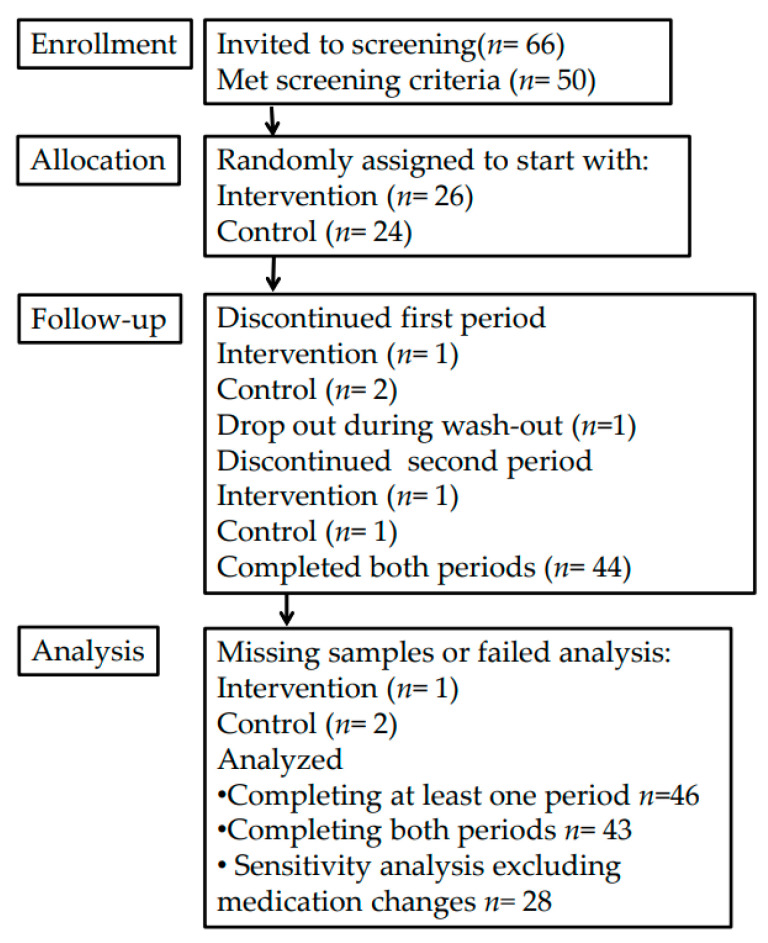
Consolidated Standards of Reporting Trials (CONSORT) diagram.

**Figure 2 metabolites-11-00632-f002:**
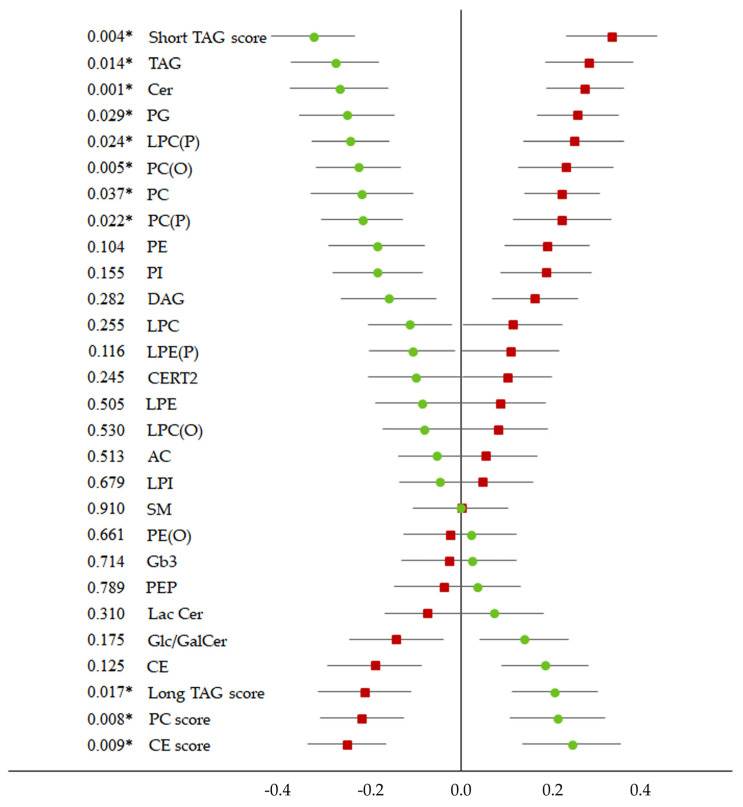
Effects of dietary intervention on lipid classes and scores associated with cardiovascular risk. Z-score standardized changes in concentrations (post–pre values) of lipid classes and scores during the intervention diet (circles) and control diet (boxes). Data presented as mean and 95% CI. P-values calculated with a mixed model. *n* = 46. * significant *p* < 0.05 AC, acylcarnitine; CE, Cholesteryl ester; Cer, Ceramide; DAG, Diacylglycerol; Gb3, globotriaosylceramide; Glc/GalCer, glucosyl/galactosyl ceramide; LacCer, Lactosylceramide; LPC, Lysophosphatidylcholine; LPC(O), Lysoalkylphosphatidylcholine; LPC(P), Lysoalkenylphosphatidylcholine; LPE, lysophosphatidylethanolamine; LPE(P), Lysoalkenylphosphatidylethanolamine; LPI, lysophosphatidylinositol; PC, Phosphatidylcholine; PC(O), Alkylphosphatidylcholine; PC(P), Alkenylphosphatidylcholine; PE, phosphatidylethanolamine; PE(O), Alkylphosphatidylethanolamine; PE(P), Alkenylphosphatidylethanolamine; PG, Phosphatidylglycerol; PI, Phosphatidylinositol; SM, Sphingomyelin; TAG, Triacylglycerol; Long TAG score (sum of TAG > 54 carbons and ≥5 double bonds; PC score (the sum of PC, LPC, PCP with ≥5 of double bonds); CE score (sum of CE > 4 double bonds); short TAG score (sum of TAG < 50 carbons and ≤4 double bonds).

**Figure 3 metabolites-11-00632-f003:**
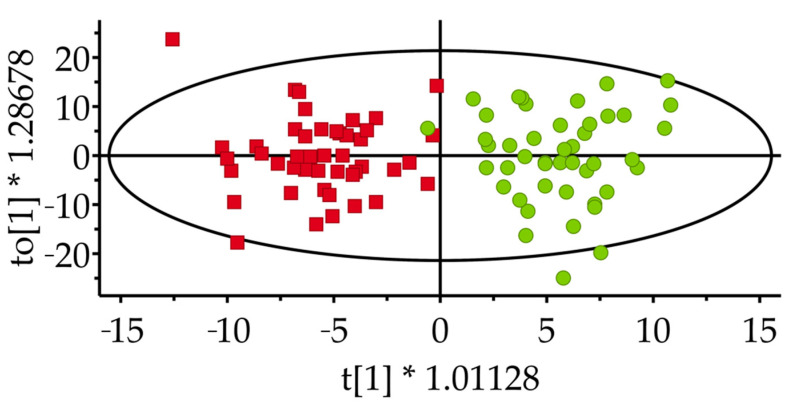
Orthogonal Projections to Latent Structures with Discriminant Analysis score scatter plot of delta values (post–pre) for the intervention diet (stars) and control diet (boxes).

**Figure 4 metabolites-11-00632-f004:**
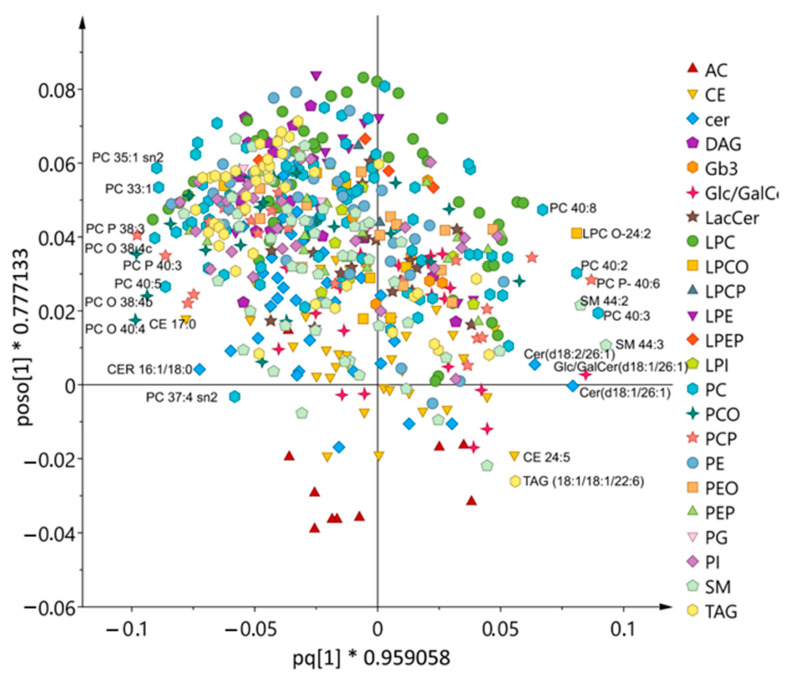
OPLS-DA loading scatter plot showing the differences in the lipid signature of the changes in serum lipid species between intervention diet (to the right) and control diet (to the left).

**Table 1 metabolites-11-00632-t001:** Baseline characteristics of participants completing at least one period (*n* = 46).

Baseline Characteristics	Median (IQR) ^c^
Sex F/M (n)	36/10
Age (year)	63 (53, 71)
Body mass index (kg/m^2^)	26.5 (24.0, 31.6)
Weight (kg)	77.2 (66.8, 84.8)
Fat mass ^a^ (kg)	27.8 (19.5, 32.3)
Lean mass ^a^ (kg)	48.2 (42.8, 52.3)
SBP (mmHg)	135 (125, 140)
DBP (mmHg)	80 (72, 89)
Serum TC (mmol/L)	5.5 (4.6, 6.0)
Serum LDL (mmol/L)	3.35 (2.78, 4.23)
Serum HDL (mmol/L)	1.60 (1.30, 2.03)
Serum triacylglycerides (mmol/L)	1.10 (0.84, 1.53)
DAS28 ^b^	3.70 (3.06, 4.65)
DAS28-CRP ^b^	3.47 (2.85, 4.14)
Medications used at baseline	*n* (%)
Cardiovascular agents	22 (48)
Vasodilator	16 (35)
Statins	7 (15)
Betablocker	7 (15)
Anticoagulants	4 (9)
Diuretics	4 (9)
bDMARD	17 (37)
csDMARD	34 (74)

TC = total cholesterol, LDL-C = low density lipoprotein, HDL = high density lipoprotein, bDMARD = biological disease-modifying antirheumatic drugs, csDMARD = conventional synthetic disease-modifying antirheumatic drugs ^a^ *n* = 42, ^b^ score, ^c^ Interquartile Range (first quartile, third quartile).

**Table 2 metabolites-11-00632-t002:** Effect of dietary intervention on CERT2, included quotas and specific lipid species.

	Intervention Pre Median (IQR) ^a^ (*n* = 46)	Intervention Post Median (IQR) ^a^ (*n* = 45)	*P* Intervention Pre vs. Post ^b^ (*n* = 45)	Control Pre Median (IQR) ^a^ (*n* = 47)	Control Post Median (IQR) ^a^ (*n* = 45)	*P* Control Pre vs. Post ^b^ (*n* = 44)	*P* I vs. C ^d^ (*n* = 46)	*P* I vs. C ^e^ (*n* = 28)
CERT2	6.0 (4.0, 8.0)	6.0 (4.0, 8.0)	0.493	6.0 (4.0, 8.0)	7.0 (4.0, 8.5)	0.041	0.245 ^f^	0.404 ^f^
CERT2 components:								
Cer(d18:1/24:1)/ Cer(d18:1/24:0)	0.56 (0.49, 0.61)	0.58 (0.47, 0.64)	0.329	0.54 (0.49, 0.61)	0.35 (0.29, 0.41)	0.082	0.135	0.182
Cer(d18:1/16:0)/ PC 16:0/22:5	0.0079 (0.0064, 0.0096)	0.0079 (0.0070, 0.010)	0.018	0.0082 (0.0066, 0.0098)	0.0087 (0.0065, 0.0097)	0.327	0.376	0.271
Cer(d18:1/18:0)/ PC 14:0/22:6	0.11 (0.084, 0.18)	0.092 (0.067, 0.14)	0.257	0.14 (0.089, 0.19)	0.12 (0.11, 0.16)	0.455	0.002	0.005
PC 16:0/16:0 ^c^	22 (19, 23)	21 (19, 24)	0.986	22 (19, 26)	22 (19, 25)	0.212	0.079	0.088
Included lipids:								
Cer(d18:1/16:0) ^c^	0.35 (0.29, 0.39)	0.34 (0.29, 0.39)	0.103	0.35 (0.30, 0.42)	0.38 (0.30, 0.42)	0.007	0.000	0.006
Cer(d18:1/18:0) ^c^	0.11 (0.081, 0.14)	0.10(0.075, 0.13)	0.291	0.11 (0.080, 0.14)	0.12 (0.088, 0.14)	0.033	0.011	0.010
Cer(d18:1/24:0) ^c^	3.2 (2.8. 3.5)	3.0 (2.6. 3.7)	0.110	3.1 (2.8. 3.6)	3.3 (2.8. 3.8)	0.054	0.005	0.002
Cer(d18:1/24:1) ^c^	1.7 (1.4, 2.0)	1.7 (1.3, 2.0)	0.243	1.7 (1.5, 2.0)	1.7 (1.4, 2.0)	0.935	0.272	0.301
PC 14:0/22:6 ^c^	0.92 (0.65, 1.1)	1.0 (0.80, 1.3)	0.013	0.88 (0.61, 1.1)	0.93 (0.70, 1.1)	0.825	0.020	0.080
PC 16:0/22:5 ^c^	43 (38, 49)	40 (35, 44)	0.000	43 (38, 46)	45 (37, 50)	0.037	0.000	0.001

CERT2, a composite score including ceramides and phospatidylcholines; I, intervention, C, control, Cer, ceramide; PC, phospatidylcholine, µmol/L, ^a^ Interquartile Range (first quartile, third quartile), ^b^ Wilcoxon signed rank test within period **^c^** µmol/L, ^d^ Mixed model, ^e^ Mixed model sensitivity analysis, ^f^ Wilcoxon signed rank test post intervention vs post control.

**Table 3 metabolites-11-00632-t003:** Models statistics.

Model	Scaling	Nr of Lv ^a^	N	R2X (cum) ^b^	R2Y (cum) ^c^	Q2 (cum) ^d^	CV-ANOVA ^e^ (*p*-Value)	Permutation Test (Q2) ^f^	Correct Classified (%C/%I) ^g^
**PCA baseline**	**UV**	7	43	0.626		0.319			
**OPLS baseline ^h^**	**UV**	1 + 1 + 0	43	0.235	0.709	0.373	0.0011		
**OPLS dietary intake ^i^**	**UV**	1 + 0 + 0	174 ^l^	0.130	0.190	0.125	<0.0001 all year		
**OPLS clinical markers ^j^**	**UV**	3 + 0 + 0	90	0.325	0.587	0.461	<0.000001 all year		
**OPLS fatty acids ^k^**	**UV**	6 + 2 + 0	90	0.521	0.794	0.659	<0.00001 all year		
**OPLS-EP**	**UVN**	1 + 1 + 0	43	0.317	0.825	0.635			
**OPLS-DA**	**UV**	1 + 2 + 0	90	0.267	0.822	0.504	5.55 × 10^−11^	−0.439	100/98
**OPLS-EP S ^m^**	**UVN**	1 + 2 + 0	28	0.408	0.961	0.697			
**OPLS-DA S ^m^**	**UV**	1 + 1 + 0	56	0.238	0.731	0.471	1.14 × 10^−6^	−0.429	86/91

PCA, principle component analysis; OPLS, Orthogonal Projections to Latent Structures; OPLS-DA, OPLS with Discriminant Analysis, OPLS-EP-OPLS with effect projections; ^a^ Latent Variables, ^b^ Cumulative fraction of the sum of squares of X explained by the selected latent variables, ^c^ Cumulative fraction of the sum of squares of Y explained by the selected latent variables, ^d^ Cumulative fraction of the sum of squares of Y predicted by the selected latent variables, estimated by cross validation, ^e^ ANalysis Of VAriance testing of Cross-Validated predictive residuals, ^f^ The intercept between real and random models, degree of overfit, ^g^ C control and I intervention, ^h^ including age, ^i^ including fibre and poly unsaturated fatty acids, ^j^ including triacylglycerides, total cholesterol, high density lipoprotein, low density lipoprotein, ^k^ including 11 of the 21 fatty acids, ^l^ data for intake missing for 6 samples ^m^ Sensitivity analysis.

**Table 4 metabolites-11-00632-t004:** Discriminating lipid species from OPLS-EP and OPLS-DA models comparing the dietary interventions.

Lipid Species	Models	Post I vs. Post C	*P* Intervention Pre vs. Post ^a^ (*n* = 45)	*P* Control Pre vs. Post ^a^ (*n* = 44)	*P* Intervention vs. Control ^b^ (*n* = 46)
PC 35:1_sn2	OPLS-DA	↓	0.005	0.000	0.000
PC(O) 34:0	OPLS-DA S	↓	0.000	0.001	0.000
PC(O) 38:3	OPLS-DA, OPLS-DA S	↓	0.000	0.000	0.000
PC(O) 38:4b	OPLS-DA	↓	0.000	0.007	0.000
PC(O) 38:4c	OPLS-DA, OPLS-DA S	↓	0.000	0.046	0.000
PC(O) 40:4	OPLS-DA, OPLS-DA S, OPLS-EP	↓	0.000	0.002	0.000
PC(P) 38:3	OPLS-DA, OPLS-DA S	↓	0.000	0.041	0.000
PC(P) 40:3	OPLS-DA	↓	0.000	0.017	0.000
PC(P) 40:6	OPLS-DA, OPLS-DA S	↑	0.000	0.000	0.000
Glc/GalCer(d18:1/26:1)	OPLS-DA	↑	0.000	0.004	0.000
SM 44:3	OPLS-DA	↑	0.000	0.002	0.000

I, intervention; C, control; PC, Phosphatidylcholine; PC(O), Alkylphosphatidylcholine; PC(P), Alkenylphosphatidylcholine; Glc/GalCer, glucosyl/galactosyl ceramide; SM, sphingomyelin. Lipids were selected if among top 20 VIP or w > 0.09. ^a^ Wicoxon signed rank test comparing pre and post values within each period. ^b^ Mixed model to compare periods.

## Data Availability

Data cannot be shared publicity because of Swedish law. The datasets analyzed in the current study are available from the corresponding author on reasonable request.

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
