# Peer review of "A Randomized Controlled Dietary Intervention Improved the Serum Lipid Signature towards a Less Atherogenic Profile in Patients with Rheumatoid Arthritis"

_metabolites, 2021, doi:10.3390/metabo11090632_

Round 1

Reviewer 1 Report

In the study entitled: „A randomized controlled dietary intervention improved the serum lipid signature towards a less atherogenic profile in patients with rheumatoid arthritis” the effect of diet on lipid composition in serum was investigated. Subjects were patients with rheumatoid arthritis. Size of the study group is moderate but sufficient to get reliable results.

The information of antirheumatic drugs used by patients is lacking in the presented version of the manuscript. The detailed information is needed. Moreover, the appropriate statistical analysis will be welcomed. Consequently, the paragraph presenting the effect of antirheumatic drugs on lipids should be implemented in discussion section.  

Author Response

We thank the reviewer for this comment; this is obviously important information that should be included. All medication of importance and that could influence serum lipids are now added in Table 1.We have also added a short discussion about the pharmacological treatment and what influence it may have on the results (Row 322-331). Because we in advance knew that the patients would have different medication and that this medication could influence the outcome, we choose a cross-over design. Therefore, the common problem with differences between groups (in for example medication) is not an issue and thus there is no need to adjust for the medication in the statistics analysis. However, it is of course important to know what medication the patients were on as a background information and we also added a sensitivity analysis excluding periods with changes in medication, to make sure that these did not have an impact on the results.

Reviewer 2 Report

Thank you for the opportunity to review this work.  This is a very interesting study into the impact of diet on the serum lipids in patients with rheumatoid arthritis.  The results of this study support the potential role of dietary modification in atherogenic risk reduction in this at-risk population.  It is a well written descriptive analysis of the findings.

As stated in the review, the manuscript was investigating the effect of a dietary intervention on the serum lipid profile in a cohort of RA patients using a cross-over design.  As a result of the well established increase in risk of CV disease in patients with RA, the impact of diet on the atherogenic risk serum lipid profile is of high relevant and interest.

The topic overall isn’t novel and represents a focus of research into the atherogenic risk associated with RA and other autoimmune conditions.  Although it may not be considered novel, it is still an important contribution to the literature on the topic. This work specifically addressed the lipid profiles, applying the CERT2 phospholipid-based CV disease risk score.  Although the CERT2 score wasn’t significantly altered by the dietary intervention, there was a correction in a number of the lipid classes associated with atherogenic risk.   

The paper is well written and easy to read. The authors conclude that the dietary intervention failed to have a significant impact on the CERT2 CVD risk score, but noted an improvement in the overall serum lipid profile with the dietary intervention.

Author Response

Thank you very much for this very positive and nice review of our paper.